**Data Availability Statement:** All relevant data are within the paper and its Supporting Information files.

**Funding:** The authors received no specific funding for this work

# 'We dry contaminated meat to make it safe': An assessment of knowledge, attitude and practices on anthrax during an outbreak, Kisumu, Kenya, 2019

**Bernard Chege Mugo** *, **Cornelius Lekopien, Maurice Owiny**

Field Epidemiology and Laboratory Training Program, Ministry of Health, Nairobi, Kenya

* bernardchegevet@gmail.com

## Abstract

### Introduction

Anthrax is the highest-ranked priority zoonotic disease in Kenya with about ten human cases annually. Anthrax outbreak was reported in Kisumu East Sub County after some villagers slaughtered and ate beef from a cow suspected to have died of anthrax. We aimed at establishing the magnitude of the outbreak, described associated factors, and assessed community knowledge, attitude, and practices on anthrax.

### Methods

We reviewed human and animal records, conducted case search and contact tracing using standard case definitions in the period from July 1 through to July 28, 2019. A cross-sectional study was conducted to assess community knowledge, attitude, and practices towards anthrax. The household selection was done using multistage sampling. We cleaned and analyzed data in Ms. Excel and Epi Info. Descriptive statistics were carried out for continuous and categorical variables while analytical statistics for the association between dependent and independent variables were calculated.

### Results

Out of 53 persons exposed through consumption or contact with suspicious beef, 23 cases (confirmed: 1, probable: 4, suspected: 18) were reviewed. The proportion of females was 52.17% (12/23), median age 13.5 years and range 45 years. The attack rate was 43.4% (23/53) and the case fatality rate was 4.35% (1/23). Knowledge level, determined by dividing those considered to be 'having good knowledge' on anthrax (numerator) by the total number of respondents (denominator) in the population regarding cause, transmission, symptoms and prevention was 51% for human anthrax and 52% for animal anthrax. Having good knowledge on anthrax was associated with rural residence [OR = 5.5 (95% CI 2.1–14.4; p<0.001)], having seen a case of anthrax [OR = 6.2 (95% CI 2.8–14.2; p<0.001)] and among those who present cattle for vaccination [OR = 2.6 (95% CI 1.2–5.6; p = 0.02)].

**Competing interests:** The authors have declared that no competing interests exist

About 23.2% (26/112) would slaughter and sell beef to neighbors while 63.4% (71/112) would bury or burn the carcass. Nearly 93.8% (105/112) believed vaccination prevents anthrax. However, 5.4% (62/112) present livestock for vaccination.

## Conclusion

Most anthrax exposures were through meat consumption. Poor knowledge of the disease might hamper prevention and control efforts.

## Introduction

Anthrax is a zoonotic disease of public health significance, associated with human and livestock morbidity and mortality as well as economic losses due to decreased trade in livestock and derived produce due to prohibition of movement of animals and products of animal origin during quarantine. The disease is caused by a gram-positive, spore-forming rod-shaped bacteria, *Bacillus anthracis (B.anthracis)*. A carcass of previously infected animals can contaminate soil thus making the contaminated soil serve as a natural reservoir for anthrax spores [1]. Susceptible animals get infected while grazing in areas contaminated with anthrax spores resulting in a cycle of infection, death, and release of spores that could contaminate new areas [1,2]. Transmission of the disease to humans mainly occurs through contact or consumption of infected animal carcasses and contaminated animal products. Anthrax in humans is classified into three forms depending on the route of transmission and clinical picture; Cutaneous form, inhalation form (respiratory), and ingestion form [3]. Cutaneous anthrax is the most common form, accounting for up to 95% of all cases. Inhalation form produces the most severe human disease followed by ingestion form (oropharyngeal or gastrointestinal). However, all three forms have the potential to progress to a fatal systemic infection. The incubation period of anthrax in humans is 1–7 days, with a median of three days dependent on the form of anthrax; cutaneous form is 2–3 days, pulmonary form is varied and can get delayed up to one or 2 months, and gastrointestinal form is 2–5 days but could be as short as 15 hours [2]. The disease occurs as sudden death in livestock while in humans, clinical symptoms are non-specific and dependent on the route of entry of spores into a susceptible host [4]. Symptoms in Human eschars, headache, fever, nausea, vomiting, abdominal pain, edema, and chest pain. The disease is an occupational hazard for farmers, shepherds, butchers, leather works, bone and wool processors, wildlife attendants, and veterinarians. Individuals with cuts and abrasions are at high risk of infection when handling infected carcasses [5]. Carnivores e.g. dogs and omnivores e.g. pigs and are more resistant to anthrax compared to herbivores and therefore are agents of spread and environmental contamination with *B. anthracis* spores by their ability to carry *B. anthracis* spores in fomites to new areas [6].

Most countries, including Kenya, consider anthrax as a notifiable disease with one case of anthrax being considered as an outbreak [7]. The seasonality of anthrax varies among locations making it difficult to develop a single consistent ecological description of anthrax [8–10]. Most outbreaks occur late in the dry seasons and at the end of heavy rains suggesting that extreme weather may be an important trigger of anthrax outbreaks [11]. Cases of anthrax in humans and livestock in developed countries are few and sporadic. The disease is enzootic in parts of Africa, the Middle East and Central Asia Outbreaks in wildlife have been reported in North America, Europe, and Sub-Saharan Africa [12,13]. In Kenya, anthrax is the highest-ranked priority zoonotic disease based on the burden, social-economic impact, and severity of the

disease; on average, ten (10) anthrax cases occur annually at the human-animal interface based on data from medical and veterinary records [14]. The number of cases could be higher than this with a possibility of undocumented cases due to underreporting. The national human seroprevalence survey in 2017 reported seropositivity of 11.3% with some regions reporting up to 28% seropositivity [15]. Outbreaks of anthrax in animals and humans have been reported in Murang'a, Nakuru, Bomet, Meru, and Narok counties with the disease considered endemic in these counties due to high incidences. Reports of anthrax outbreaks in Kisumu county are very few according to reports found at the office of the director of veterinary services and therefore considered a low-risk county.

On July 11, 2019, the Kisumu County Department of Health reported a confirmed case of human anthrax of a 32-year-old man from Kisumu East Sub County. Confirmation was done by collecting a whole blood sample and demonstrating *B. Anthracis* on a gram-stained blood smear microscopic slide. This man had taken an active role in the slaughter of a cow that had died of suspected anthrax on July 5, 2019. The patient had been referred to Kisumu County Referral Hospital (KCRH) on July 9, 2020, presenting with extreme pain, fever, edema of the right arm, and died on July 11, 2020. On July 12, 2019, four other patients were admitted to the facility after which they stabilized and were discharged. Several patients from Mowlem village presented themselves for a medical checkup at KCRH and the nearby Nyalunya Health Center. Those who presented themselves for medical checkups had either consumed beef or participated in the slaughter of the suspect cow. Most patients presented with fever, body pain, and general weakness. The existing knowledge and attitude towards pathogen infections in the community influences practices that possibly enhance risks for anthrax infection and outbreak. Assessment of the level of knowledge regarding anthrax in the community is suitably done by conducting Knowledge, attitude and practice studies (KAP) [16–18].

We investigated this outbreak to determine its magnitude, assessed community knowledge, attitude, and practices including associated factors, and formulate the most appropriate measures for control of future anthrax outbreaks in Kisumu East Sub County. Before the outbreak, there were no activities regarding the control of anthrax. However, during the outbreak, the department of health did disinfection of the contaminated areas and community sensitization on anthrax.

## Methods

### Investigation site

Kisumu East Sub County is one of the six sub-counties in Kisumu County (longitudes 33˚ 20'E and 35˚ 20'E and latitudes 0˚ 20'South and 0˚ 50'South). T. It has five wards namely Kajulu, Kolwa central, Kolwa east, Manyatta B, and Nyalenda A [19]. The Sub County had a total population of 220,997 persons with 61,499 households and an area of 142 sq. kilometers as per the 2019 Census [20]. Approximately 46% of its population live in the Periurban while the rest 54% is settled in the rural areas. The main economic activities are crop farming and livestock keeping practiced in Kolwa East, Kolwa West & Kajulu Wards. Residents of Manyatta B and Nyalenda A engage in small-scale business enterprises, residential housing, and livestock keeping. There have been previous reports on anthrax outbreak in area according to reports from the director of veterinary services. However, none of those reports has been documented.

### Case definition and identification

Anthrax occurs, as an outbreak and one confirmed case constitutes an outbreak. We conducted the outbreak investigation for a period of eight days from July 17, 2019 through July 24, 2019, and used the case definition for human anthrax. A suspected case was defined as a

person with an acute illness presenting with any one of the following signs and symptoms; fever, hand swelling, small skin blisters, skin sores with a black centre, dyspnoea, chest-pain, stomach-ache, headache, nausea, fatigue, and dizziness from July 1 through July 13, 2019, residing at Mowlem village, Kisumu East Sub County. A Probable case was a suspected person with an epidemiological link to a confirmed livestock anthrax case having participated in the slaughtering of the dead cow confirmed of anthrax. A Confirmed case was a suspected person with a blood sample positive for *B. anthracis* on microscopy.

## Data collection

We did a review of inpatient and outpatient records at Kisumu County Referral Hospital (KCRH) and Nyalunya Health Centre using the case definition to identify cases. We used a standardized abstraction tool to collect patient data on age, sex, place of residence, facility name, date of onset, date of admission, patient status, and presenting signs and symptoms. We then searched the patient's history for information on whether the patients had a confirmed laboratory diagnosis, participated in the slaughter or consumption of beef from the suspected cow.

The county health team and community health workers supported us to conduct an active case search and contact tracing by identifying the residence of the index case. Using a snow-balling sampling technique, we identified other persons in the village who came into contact with the infected carcass by either handling, cooking, consumption, slaughtering, touching blood or fluids, preservation of the hide, and disposal of ingesta from the suspected cow. We searched for patients with cutaneous lesions consistent with anthrax infection and conducted a verbal autopsy by interviewing the spouse of the deceased.

To assess the community knowledge, attitude and practices (KAP) on anthrax, we conducted a cross-sectional study. We calculated the sample size using Cochran exact formula [21] and used multistage sampling to select the households that were recruited in the study. In the first stage of sampling, all the five wards in Kisumu east Sub County were purposefully selected to achieve a geographical representation.

The sampling unit was a household. We calculated sample size using Cochran formula with 96% proportion for awareness (UON Repository) [16]

$n = Z^{2*} P (1-P)/d^2$

Where;

Z-value = 1.96 (Standard normal deviate for the 95% significance level)

p = Prevalence (96%) Proportion of people aware of anthrax

d = precision (0.05 at 95% significance level)

Therefore, n = {$1.96^2 * 0.96(1–0.96)/0.05^2$} = 56

Adjusting the sample size for design effect; 56*2 = 112 Households

This sample size was considered to adequately represent the larger population.

In the second stage of sampling, the total number of households to be sampled (112) were allocated proportionally to households in each ward and then adjusted according to livestock density by ward. To determine the households to be visited, the team identified a central reference point in each ward which was a school, Health facility, church, Shopping center, or government office. From the reference point, we spun a bottle and identified the direction of the bottle top. We selected the first household in that direction and interviewed every 5th household until half of the households allocated for that ward were interviewed. We then went back to the reference point and moved in the opposite direction, interviewed the first household, and then every 5th household until we completed all the households allocated for that ward. We did the same in all the wards until we interviewed all 112 households.

At the household level, we administered consent and interviewed the head of the household Using a standardized electronic questionnaire in Epi-Info. In the absence of the household head, the next senior member of the household aged 18 years or older was interviewed. We collected demographic information of persons being interviewed including age sex place of residence occupation, religion. Additionally, we collected data on types of animals kept, knowledge on the cause, transmission, identification by signs and symptoms, and prevention for anthrax. We also collected data to assess practices including animal slaughter, sale, and carcass disposal. Data on attitude included animal slaughter, consumption, and control of anthrax. Reasons for the various responses were recorded as qualitative data for attitude and practices.

## Inclusion and exclusion criteria

The study inclusion criteria were households keeping livestock in the selected villages and those whose household head or any person above the age of 18 years was available. The exclusion criteria for the household survey were those households in which the respondents were eligible for the interview but could not be able to provide the information required due to the inability to communicate.

## Data management and analysis

Data from outpatient registers at the health facilities in the Sub County and county referral hospital was abstracted using a standard tool, entered, cleaned, and analyzed using Microsoft Excel (Microsoft Office, Seattle, USA) and Epi Info 7 (CDC, Atlanta, Georgia, USA). The KAP data from electronic questionnaires were downloaded, cleaned, and analyzed.

We described cases by person, place, time, and clinical presentation, calculated case fatality rate, attack rates (AR) by age and sex using the total number of at-risk by exposure through slaughtering the cow, handling, cooking, and eating the suspicious beef. To describe the progression of the outbreak, we constructed an epidemic curve and used it to determine the minimum, median, and maximum incubation period of the disease.

We assessed the level of knowledge of participants on human and animal anthrax by asking questions regarding cause, transmission, clinical signs and symptoms, and prevention. In scoring the level of knowledge, one mark was awarded for the correct response and zero for the wrong response. We determined the proportion of questions answered correctly The knowledge scale was adopted from Traxler RM et al by summing together the scores (0 = No, or 1 = Yes) from the set of questions regarding the respondents' knowledge of anthrax in animals and humans [22]. We calculated total scores for all the respondents and determined the mean total score. A higher score than the mean indicated a greater overall knowledge of anthrax Respondents with a score above the average total score were considered to 'have good knowledge' while those with a score below the mean were considered to 'have poor knowledge' on anthrax. Knowledge level was determined by dividing those considered to be 'having good knowledge' on anthrax (numerator) by the total number of respondents (denominator). The difference in the means of level of knowledge on human and animal anthrax was tested for statistical significance using a paired t-test.

The responses were grouped into those who have good knowledge and those with poor knowledge of anthrax. Using the level of knowledge as a dependent variable and residence, having seen anthrax, cattle vaccination, sex, education, occupation as independent variables, we determined the association between the dependent and independent variables. At bivariate analysis, we calculated Odds Ratios (OR) and their corresponding 95% confidence interval (95% CI) for the association between independent and dependent variables. We conducted a Chi-squared test to determine the statistical significance of the association We then conducted

a multivariate analysis using backward elimination logistic regression for variables that had p-values < 0.2 at bivariate analysis. Factors whose association had p-values < 0.05 at multivariate analysis were considered to be independently associated with 'having good knowledge' on anthrax. Qualitative data were was analyzed thematically.

### Ethical considerations

This outbreak being an acute public health problem, the investigation did not require approval by the institutional review board. Approval to conduct the study was sought from the Ministry of Health and Ministry of Agriculture, Livestock and Fisheries/Directorate of Veterinary Services, and the department of health services of the county government of Kisumu. We administered verbal consent to eligible study participants and the confidentiality of the information from the participants was maintained through the use of codes and password-protected computers and databases.

## Results

### Description of cases

We identified 23 cases out of 53 persons who were exposed to the suspicious beef. Of the cases, 18 were suspected, four probable and one confirmed by demonstration of *B. Anthracis* using microscopy in a gram-stained blood smear. The mean age of cases was 21 years (SD = ±14). The majority of cases were aged between 5 and 12 years at 16.9% (9/12) (Table 1). Of the total number of cases identified, 43.5% (10/23) were referrals to KCRH based on the severity of signs and symptoms (Table 1). The majority of these referrals belong to the age group between 17 and 35 years at 60% (6/10). The overall attack rate was 43.4% (23/53), with the highest recorded among ages between 17 and 35 years at 62.5% (5/8), and males at 44.3% (12/27) (Table 1). Among the presenting signs and symptoms were fever at 41.7% (10/23) and oedema at 37.5% (10/23) (Table 2).

Verbal auto revealed that the index case was a livestock trader and that he got sick on July 6, 2019, a day after participating in the slaughter of a cow suspected to have died of anthrax. On July 8, 2020, he was treated for allergic reactions in a private clinic but his condition deteriorated with increasing body swelling and pain. The number of cases gradually increased, peaked on July 11, 2020, and ended on July 12, 2020 (Fig 1). Kisumu county department of health conducted active case finding, contact tracing while facilitating referrals and disinfection but the veterinary authorities were not engaged in the exercise. The minimum incubation period was

**Table 1. Frequency distribution, facility, age, gender-specific attack rates for Anthrax Kisumu East Sub County 2019.**

| Variable | Frequency | Proportion (%) | Attack rates (%) |
|---|---|---|---|
| **Age group** | | | |
| 5−12 | 9 | 16.9 | 36.0 |
| 13−16 | 5 | 9.4 | 41.7 |
| 17−35 | 5 | 9.4 | 62.5 |
| 36−64 | 4 | 7.6 | 50.0 |
| **Gender** | | | |
| Male | 12 | 22.6 | 42.3 |
| Female | 11 | 20.8 | 44.4 |
| **Health Facility** | | | |
| KCRH | 10 | 43.5 | 100.0 |
| Nyalunya HC | 13 | 56.5 | 30.2 |

**Table 2. Frequency distribution of anthrax symptoms, Kisumu East Sub county 2019 (n = 23).**

| Variable | Frequency | Proportion (%) |
| --- | --- | --- |
| **Symptoms** | | |
| Fever | 10 | 41.7 |
| Oedema or Skin lesions each | 9 | 37.5 |
| Dyspnoea or Chest pain each | 8 | 33.3 |
| Stomach ache | 7 | 29.2 |
| Headache | 3 | 12.5 |

Nausea, Fatigue and Dizziness each had a frequency of 1.

1 day with a median of 7days. There were no reported signs in the suspected cow before death since it died suddenly. However, blood was reported to be oozing from body orifices after death.

## KAP study results

Respondents for the KAP study were 112, with 51.8% (58/112) being female. The mean age was 52.5 years (SD = 17.9 years) with a majority, 76.8% (86/112) being aged above 35 years. Most respondents, 88.4% (99/112) had acquired formal education, farmers were 58% (65/112) while the most common livestock kept was cattle at 95.5% (107/112).

   **Knowledge.**   Of the 400 questions asked on the cause, transmission, clinical symptoms, and prevention in human anthrax, 51% (204/400) were answered correctly by respondents while regarding while that of anthrax in animals was 52% (208/400) (Table 3). Statistical test of

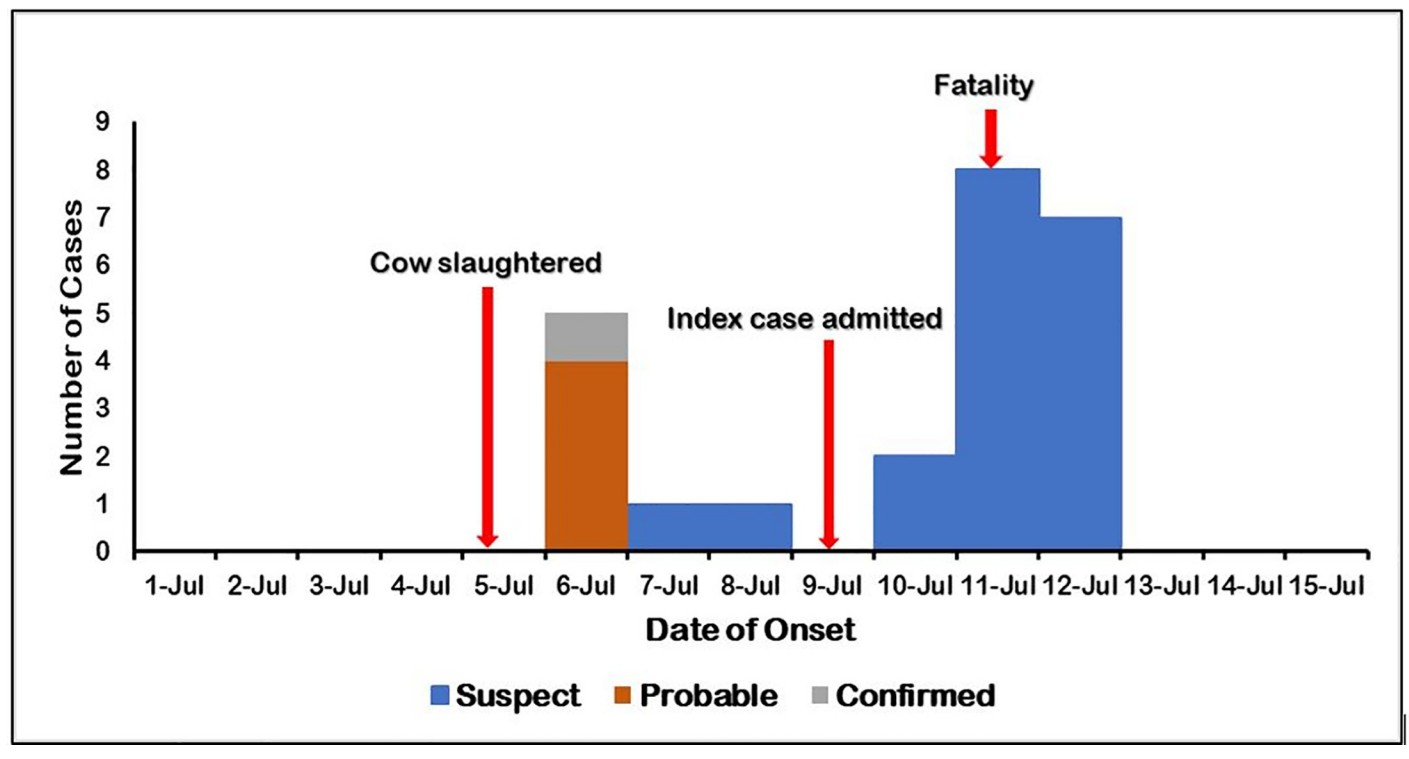

**Fig 1. Epidemic curve for anthrax outbreak, Kisumu East Sub County, July 2019.**

**Table 3. Knowledge level for human and animal anthrax, Kisumu east Sub County, July 2019.**

| Variable (n = 112) | Knowledge Level(Human) | | Knowledge Level(Animals) | |
|---|---|---|---|---|
| | Frequency | Proportion (%) | Frequency | Proportion (%) |
| Cause | 32 | 28.6 | 32 | 28.6 |
| Transmission | 91 | 81.3 | 51 | 45.5 |
| Humans symptoms | 39 | 34.8 | 59 | 52.8 |
| Prevention | 66 | 58.9 | 89 | 79.4 |
| Knowledge level (Mean) | | **51** | | **52** |

significance in the difference between the two means revealed a p-value of 0.96 at the 95% level of confidence among participants One of the elderly male respondents from Nyalenda village explained his knowledge of Anthrax by saying, *"I saw a disease of cattle in rift valley when we were young before my parents moved here. Healthy cows grazing in the fields died suddenly with bleeding from the mouth, nostrils and had distended belly, we called it 'Aremo' (bleeding). We have not seen it here and the young generation may not know about it"*.

**Practices.** On the assessment of practices towards sudden death of an animal, we found out that 23.2% (26/112) would slaughter and sell meat to neighbors, 2.7% (3/112) would remove skin and burry, 11.6% (13/112) would remove skin and feed meat to dogs. About 5.4% (6/112) would seek post-mortem services and 63.4% (71/112) would bury or burn the carcass. However, no one would sell meat to the butchery; instead, they would preserve the meat by drying it for two weeks (to make a type of meat product called *Aliya* in local dialect) and believe it to be safe for human consumption after the drying period. A female respondent at Chiga village, East Kolwa ward, raised this while explaining, *"When a cow dies and we do not know the cause of its death, we normally slaughter the dead cow and share meat with neighbors. Meat is cut into long thin slices and placed on iron sheet roof for 2 weeks to dry completely, we call it Aliya and it is safe to cook and eat"*.

**Attitude.** An assessment of practices contributing to the transmission of anthrax revealed that 15.2% (17/112) of the respondents would consume meat from a dead animal, 12.5% (14/112) would skin an animal that died suddenly, 52.7% (59/112) would conduct home slaughter and 11.6% (13/112) would consume uninspected meat. A male respondent at Ogandi village in Kajulu ward explained the reasons for these practices by saying the following;

*"We eat meat from a dead cow since it is sold at a cheaper price than meat from the butchery and the owner can allow you buy on credit to pay later provided he/she gets back some little value in return for the cow. The owner also skins the cow so that in case the meat cannot be eaten, he/she can sell the skin since it is always healthy"*.

Those who believed that livestock vaccination could prevent anthrax were 93.8% (105/112) but those who would vaccinate cattle against anthrax were 55.4% (62/112). When asked what could influence their turn up at vaccination centers 4.5% (5/112) stated high cost, 2.7% (3/112) stated time consumption and 2.7%(3/112) stated long distances to vaccination sites as contributory factors to poor turn up at vaccination center. A Majority, 62.5%, (70/112) of respondents would engage animal health service providers in the management of livestock diseases while those who considered anthrax to be a serious disease in livestock were 86.6% (97/112).

A female respondent from Kalucu village in Kajulu ward explained, *"Any disease that causes bleeding and death like anthrax is dangerous to our animals and should be vaccinated. The problem is animals are vaccinated in a fixed crush, we have to walk our animals there and we have*

**Table 4. Factors associated with having knowledge regarding anthrax, Kisumu East Sub County, July 2019 (n = 112).**

| Variable | Variable description Frequency | | Odds Ratio | 95% Confidence Interval | P-value |
|---|---|---|---|---|---|
| Having seen Anthrax | Yes | 57 | 6.2 | 2.8–14.2 | <0.001 |
| | No | 55 | | | |
| Residence | Rural | 81 | 5.5 | 2.1–14.4 | <0.001 |
| | Periurban | 31 | | | |
| Vaccinate cattle | Yes | 62 | 2.6 | 1.2–5.6 | 0.02 |
| | No | 50 | | | |
| Sex | Male | 54 | 1.7 | 0.8–3.5 | 0.25 |
| | Female | 58 | | | |
| Education | Educated | 99 | 1.2 | 0.4–4.0 | 0.95 |
| | Not educated | 13 | | | |
| Occupation | Farmer | 82 | 0.9 | 0.4–2.0 | 0.92 |
| | Not farmer | 30 | | | |

*other work to do. Therefore, if you are far from the crush it is not possible to bring animals for vaccination. Vaccinators should be moving from one house to house"*

**Factors associated with having knowledge.** On bivariate analysis we found residence (OR = 5.5, CI: 2.1–14.4), having seen anthrax case (OR = 6.2, CI: 2.8–14.2) and presenting cattle for vaccination (OR = 2.6, CI: 1.2–5.6) to be associated with knowledge on anthrax regarding cause, transmission, symptoms, and prevention (Table 4). On multivariate analysis, residence (adjusted OR = 4.2, CI: 1.4–12.5) and having seen anthrax (adjusted OR = 5.6, CI: 2.4–13.4) were independently associated with knowledge regarding anthrax (Table 5).

## Discussion

This outbreak investigation documented occurrence of cutaneous and gastrointestinal forms of anthrax among people who either consumed beef or participated in the slaughter of a cow suspected to have died of anthrax in Kisumu East Sub County. The presenting clinical symptoms including oedema and skin lesions are evidence of a cutaneous form of anthrax. Stomach ache and nausea are indicative of gastrointestinal form while the rest of the symptoms are a result of systemic involvement most likely associated with consumption of the suspicious beef pointing to gastrointestinal form. Demonstration of *B. anthracis* in a blood sample from the index case who took an active role in the slaughter of a cow suspected to have died of anthrax and reports of additional cases linked to the index case by consumption of beef or contact with carcasses of infected cow provided evidence for anthrax outbreak [23]. Although cases of anthrax have been documented in counties considered endemic like Murang'a, Nakuru, Bomet, and Narok [24] incidences of anthrax in humans or animals are very few in Kisumu East Sub County which is considered a low-risk area. The level of knowledge on anthrax could

**Table 5. Factors independently associated with having knowledge regarding anthrax, Kisumu East Sub County, July 2019.**

| Variable | aOR | CI | P-value |
|---|---|---|---|
| Having seen anthrax symptoms | 5.6 | 2.4–13.4 | 0.0001 |
| Rural residence | 4.2 | 1.4–12.5 | 0.009 |
| Presenting cattle for vaccination | 1.5 | 0.6–3.7 | 0.3931 |

aOR: Adjusted Odds Ratio, CI: 95% Confidence Interval.

be considered low from our findings probably because there are very few outbreaks of anthrax in the area. Low knowledge level of the disease in the area could be the reason anthrax is excluded from the list of differentials for human cases. This exclusion could result in a delayed confirmation, misdiagnosis, and wrong treatment as it happened with the index case in this outbreak, with a potential to cause progression to severe disease and fatality [25].

The decision for Referral of a patient to KCRH was based on the degree of severity of symptoms rather than suspicion of anthrax, with most referrals being persons who participated in the slaughter of the suspected cow including the index case. The difference in the severity of infection between those referred to KCRH and those treated at Nyalenda Health Center a local health facility could be attributed to the intensity and duration of exposure to infecting bacteria [26]. The highest attack rate was observed among persons aged between 17 to 35 years. The majority of the referrals to KCRH belong to this age group. This age group is usually sought for as casual labourers and may have been engaged in the slaughter of the suspected cow, thus increased the risk of their exposure to *B. anthracis*. This group is also likely to be responsible for the preparation of meals at the household level further increasing the risk of exposure in case the beef is from a cow infected with anthrax.

The chronology of events during this outbreak indicated a delay in detection, diagnosis and management of anthrax. This could be due to poor collaboration between the department of health and veterinary authorities in the area. Employing a One Health approach during investigations of outbreaks of zoonotic diseases like anthrax could go a long way in early detection, control and management of human anthrax cases as was evidenced in a study done in Tanzania [27]. Most zoonotic diseases occur in animals before crossing to humans, thus the collaboration between the animal health service providers, who are normally the first to be notified of the occurrence of zoonotic diseases in animals and the department of health could shorten the period between diagnosis and initiation of treatment to control and contain the disease outbreaks. Vaccination of animals to control the disease at the animal level and community sensitization through risk communication are additional desirable aspects of one health approach to the problem. Anthrax outbreak occurred in July in the dry season after a period of prolonged rainfall. This is consistent with a study that reported two-thirds of anthrax outbreaks in wildlife occurred during seven months of the dry season including July [28]. Since anthrax is soil-borne, the occurrence of outbreaks at the beginning of a dry season could be attributed to reduced pastures resulting in herbivores eating grass too close to the ground where spores may be found. Stress factors associated with dry seasons including poor nutrition could lower $LD_{50}$ for *B. Anthracis* toxins thus increasing susceptibility to infection.

There were reports of anthrax outbreaks in the investigation area in the 19[th] century and this could explain the possibility of the existence of *B. Anthracis* in the area since anthrax spores have been shown to remain viable in the soil for many years [29]. However, there were possibilities that the implicated animal could have been brought into the area from an endemic zone as part of livestock trade or the owner of the animal may have visited an infected farm then carried the anthrax spores in fomites to the farm since he was a livestock trader according to narrations from the verbal autopsy.

Low level of knowledge on anthrax among residents of Kisumu East Sub County, having no significant statistical difference in knowledge between human and animal anthrax could be attributable to the probable low incidence of anthrax in the area. Association of knowledge on anthrax with having seen a case of anthrax and residing in rural areas could be attributed to a high population of livestock and intensive livestock activities in the area [10]. Anthrax in animals presents with characteristic symptoms, which include sudden death, unclotted blood and absence of rigor mortis. These symptoms are very easy to recall for one who has seen an animal anthrax case in the past. There are no differences in gender, education level and occupation

variations between the rural and peri-urban settlement resulting in a lack of association of these dependent variables with knowledge of anthrax. Engagement of the community in surveillance activities could be vital in increasing knowledge of the epidemiology of the disease and early reporting of any unexplained sudden deaths of animals, and the presence of signs consistent with anthrax in humans [3].

The existence of community attitudes and practices predisposing to anthrax infection including slaughter and consumption of dead animals could be attributed to the desire to salvage the animal. Limited knowledge on risk to consumers upon consumption of such beef is also a possibility [30]. Another notable finding from the community was the community belief that dried meat is safe from food-borne pathogens like bacteria. The community therefore dried meat by placing it on an iron sheet roof during the day for two weeks without any other form of treatment before consumption. This illusion that meat could be safe after drying may put people at risk of infections like anthrax through consumption of beef from an animal that dies suddenly from such infections. A study revealed that dried foods are not inherently safe microbiologically and may require additional treatment to achieve microbial levels that could be considered safe [31]. However, this is not the case for anthrax due to the ability of *B. anthracis* to sporulate with spores becoming more resistant to any form of treatment. Therefore, drying of anthrax-contaminated meat may not guarantee its safety for human consumption since *B. anthracis* spores may stay in dried meat and on surfaces used for drying for a long time leading to the possibility of aerosol transmission during the preparation of the dried meat or from the fomites. Additionally, drying may contaminate the soil or pasture for livestock if the meat is left to dry in the open. A study elaborated how aerosol transmission of anthrax spores could occur from dried animal by-products like hides used in making the African drums [32]. Further studies could elucidate the viability of anthrax spores from dried meat and unanticipated untoward gastrointestinal signs associated with consumption of such dried meat depending on the mode and duration of drying.

## Limitations

During this investigation, we could not obtain blood samples, body fluid, or beef samples from the suspected cow for laboratory confirmation of *B. anthracis* as recommended by guidelines [33]. In addition, we neither interviewed the incarcerated owner of the cow nor his family members to help us determine the possible source of the suspected cow to enable traceback. Although *B. anthracis* was demonstrated in blood samples obtained from the human index case, we could not link its source directly to the suspect cow.

## Conclusion

There was poor knowledge of anthrax among residents of the study area probably due to low incidences of the disease. Most people became exposed through the consumption of suspected beef. The risk of infection in a community with poor knowledge of the disease, misdiagnosis, and delayed treatment could contribute to the escalation of the magnitude of the disease outbreak. This situation could become worse with community attitudes and practices that can potentially support the transmission of anthrax.

## Recommendations

We recommended the enhancement of community sensitization on anthrax through radios and workshops to increase the level of knowledge, especially regarding cause, transmission, symptoms, and prevention. Improved veterinary public health including inspection of butcheries and banning home slaughter could reduce the risk of anthrax transmission. Vaccination

coverage for livestock in both anthrax endemic and non-endemic areas should be encouraged, in addition to farmers' education on the importance of vaccination against anthrax to improve herd immunity. Livestock disease surveillance and reporting for early detection to aid control and prevention efforts should be enhanced through collaboration between human health and animal health service providers in handling zoonotic diseases. There is a need to conduct a study to ascertain the microbiological safety of dried meat for human consumption, especially in relation to *B. anthracis*.

## Supporting information

**S1 Fig. Epidemic curve for anthrax outbreak.**
(TIF)

**S1 Table. Frequency distribution of anthrax cases by facility, age, and gender-specific attack rates.**
(TIF)

**S2 Table. Frequency distribution of anthrax symptoms.**
(TIF)

**S3 Table. Knowledge level for human and animal anthrax.**
(TIF)

**S4 Table. Factors associated with having knowledge regarding anthrax.**
(TIF)

**S5 Table. Factors independently associated with having knowledge regarding anthrax.**
(TIF)

**S1 Appendix.**
(DOCX)

**S1 Dataset.**
(XLSX)

## Acknowledgments

The authors wish to thank the Kisumu County Government departments of health and agriculture for allowing the team to conduct the investigation. We are grateful to the community members who took part in the study.

## Author Contributions

**Conceptualization:** Bernard Chege Mugo, Maurice Owiny.

**Formal analysis:** Bernard Chege Mugo.

**Investigation:** Bernard Chege Mugo, Cornelius Lekopien, Maurice Owiny.

**Methodology:** Bernard Chege Mugo, Cornelius Lekopien, Maurice Owiny.

**Supervision:** Maurice Owiny.

**Validation:** Maurice Owiny.

**Writing – original draft:** Bernard Chege Mugo.

**Writing – review & editing:** Bernard Chege Mugo.

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
