## [Decision Letter · Decision Letter 0]

9 Dec 2020

PONE-D-20-34615

‘We dry contaminated meat to make it safe’: An Assessment of Knowledge, Attitude and Practices on Anthrax during an outbreak, Kisumu, Kenya, 2019

PLOS ONE

Dear Dr. Mugo,

Thank you for submitting your manuscript to PLOS ONE. After careful consideration, we feel that it has merit but does not fully meet PLOS ONE’s publication criteria as it currently stands. Therefore, we invite you to submit a revised version of the manuscript that addresses the points raised during the review process.

This paper was reviewed by a social scientist and biological scientist.  Both felt that his was an important contribution to the literature, but that there was a need to refine the use of terminology and significantly refine the level of detail provided in the results section.  The authors are invited to make these as a major revision and re-submit the paper for consideration.

We look forward to receiving your revised manuscript.

Kind regards,

Eric Fèvre

Academic Editor

PLOS ONE

Journal Requirements:

2. In your Methods section, please provide additional location information of the study area, including geographic coordinates for the data set if available.

3.We suggest you thoroughly copyedit your manuscript for language usage, spelling, and grammar. If you do not know anyone who can help you do this, you may wish to consider employing a professional scientific editing service.  

4. In your Methods section, please provide additional information about the participant recruitment method and the demographic details of your participants. Please ensure you have provided sufficient details to replicate the analyses such as: a) the recruitment date range (month and year), b) a description of any inclusion/exclusion criteria that were applied to participant recruitment, c) a table of relevant demographic details, d) a statement as to whether your sample can be considered representative of a larger population, e) a description of how participants were recruited, and f) descriptions of where participants were recruited and where the research took place.

Additional Editor Comments (if provided):

This paper was reviewed by a social scientist and biological scientist. Both felt that his was an important contribution to the literature, but that there was a need to refine the use of terminology and significantly refine the level of detail provided in the results section. The authors are invited to make these as a major revision and re-submit the paper for consideration.

Reviewers' comments:

Reviewer's Responses to Questions

**Comments to the Author**

1. Is the manuscript technically sound, and do the data support the conclusions?

Reviewer #1: Yes

Reviewer #2: Yes

2. Has the statistical analysis been performed appropriately and rigorously? 

Reviewer #1: I Don't Know

Reviewer #2: I Don't Know

3. Have the authors made all data underlying the findings in their manuscript fully available?

Reviewer #1: No

Reviewer #2: No

4. Is the manuscript presented in an intelligible fashion and written in standard English?

Reviewer #1: Yes

Reviewer #2: Yes

5. Review Comments to the Author

Reviewer #1: Thank you for the opportunity to read this interesting article. This is a very clearly written piece of work and offers a useful summary of a recent Anthrax outbreak in western Kenya. I have been invited to review this article as a social scientist who has worked in this region. My comments primarily pertain to the KAP findings and discussion. I am not an expert on Anthrax.

In terms of the background information, it would be helpful to know if the village in question is rural or peri-urban, given the vast differences in how people live in Nyalenda and Manyatta compared to other parts of Kisumu East sub-county. Given the geographical differences across the sub-county, why is it that the KAP respondents were taken from all five wards rather than just those which shared a similar make-up to the village location?

The term ‘knowledge’ and ‘lack of knowledge’ are used in ways that are problematic from a social science perspective. There are a few different issues here. First, it is not clear what questions about Anthrax were asked of respondents. Presumably these related to modes of transmission, infection and prevention? Should this questionnaire not be included as an appendix to the article? The article should make clear that participants are being questioned on ‘biomedical understandings of modes of transmission, infection, and transmission’. It may be commonplace in biomedical publications but to gloss this as ‘knowledge about Anthrax’, but for social scientists this is unhelpful and problematic, because it undervalues other kinds of knowledge and insight that people may have about Anthrax (or aremo, or other related cattle diseases/disorders) and dismisses local knowledge as irrelevant, and as mere ‘ignorance’.

In the first quotation, you where you use the example of Aremo, it is not clear if you are giving this as an example of knowledge, or ‘lack of knowledge’. What this example does prove very nicely, is the comment in your conclusion, that previous experience is very important for contructions of knowledge, partly because the symptoms are very dramatic. Rather than just using this quote as ‘an example of knowledge’, it would be better to break up the concept of knowledge and use this quote as an example of a specific way in which knowledge is obtained. This is very helpful to know from a public health perspective, because it suggests that using images of infected cattle could be a helpful tool for educating people about the disease.

On Aliya, I wondered if you could say more about the gendered dimensions of risk here. You mention the risk of anthrax remaining in the meat despite the drying process, but not the possible risk of the preparation process. Is this a female activity? The same goes for skinning the animal. Is it more like that a man would do this? Or are these unlikely to be modes of transmission?

In general the article presentations quotations as examples of a form of knowledge, an attitude or a practice, but it doesn't always make an argument with those presentations. It is not always clear enough why the particular examples have been chosen, and what message the authors want the reader to take form the presentation of these specific examples.

Personally, I do not see much difference between ‘attitudes’ and ‘practices’ as you present them, given that both sections appear to describe what people would do if they encountered meat from a dead animal, and wonder if these two sections would be better combined.

The section on vaccination was the best written of the findings section and the most useful, because it shows how knowledge is employed but is thwarted by practicalities. It therefore highlights the lived realities of knowledge about disease in a way that could inform a public health response.

In the section where you discuss factors that increase biomedical understandings of Anthrax, why not talk about the factors that don’t have any impact? For example, it is very interesting that gender does not appear to shape access to this kinds of knowledge. Why do you think this is?

In the conclusion, I would have liked to know specifically how a one health approach would have helped here specifically. Also, in what sense does the study confirm and contradict other work? Would you expect things to be similar in Sweden anyway in terms of understanding of aetiologies and engagement in modes of prevention? These comments are left hanging a little.

Reviewer #2: while the manuscript has technically presented a sound piece of scientific research, one cannot be certain that statistical analysis been performed appropriately and rigorously. The data that computed level of knowledge for the

community members interviewed on cause transmission, symptoms and prevention need to be illustrated for transparency e.g. as a supplementary text. The manuscript also requires major language edits and appropriate referencing of literature.

6. PLOS authors have the option to publish the peer review history of their article (what does this mean?). If published, this will include your full peer review and any attached files.

Reviewer #1: **Yes: **Hannah Brown

Reviewer #2: No

---

## [Author Response · Author response to Decision Letter 0]

7 Jul 2021

PONE-D-20-34615

‘We dry contaminated meat to make it safe’: An Assessment of Knowledge, Attitude and Practices on Anthrax during an outbreak, Kisumu, Kenya, 2019

PLOS ONE

PLOS ONE

Thank you for your detailed review of my manuscript that I had submitted to PLOS ONE. I have addreees the comments raised and included the following items of my revised manuscript:

• A rebuttal letter that responds to each point raised by the academic editor and reviewer(s) uploaded as a separate file labeled 'Response to Reviewers'.

• A marked-up copy of my manuscript highlighting changes made to the original version. This has been uploaded as a separate file labeled 'Revised Manuscript with Track Changes'.

• An unmarked version of your revised paper without tracked changes uploaded as a separate file labeled 'Manuscript'.

We look forward to receiving comments on the revised manuscript.

Kind regards,

Dr. Mugo

Reviewers comments and Authors response

Reviewers comment: Please ensure that your manuscript meets PLOS ONE's style requirements, including those for file naming. 

Authors response: This has been done

Reviewers comment: In your Methods section, please provide additional location information of the study area, including geographic coordinates for the data set if available.

Authors response: We have included the geographical coordinates for Kisumu county which is longitudes 33° 20’E and 35° 20’E and latitudes 0° 20’South and 0° 50’South. The coordinates for the sampling units(Households) are available in the data set provided as supporting information.

Reviewers comment: We suggest you thoroughly copyedit your manuscript for language usage, spelling, and grammar. If you do not know anyone who can help you do this, you may wish to consider employing a professional scientific editing service. 

Authors response: We did not find any professional scientific editing service, am appealing for assistance from PLOS ONE editors 

Reviewers comment: In your Methods section, please provide additional information about the participant recruitment method and the demographic details of your participants. Please ensure you have provided sufficient details to replicate the analyses such as: a) the recruitment date range (month and year), b) a description of any inclusion/exclusion criteria that were applied to participant recruitment, c) a table of relevant demographic details, d) a statement as to whether your sample can be considered representative of a larger population, e) a description of how participants were recruited, and f) descriptions of where participants were recruited and where the research took place.

Reviewers comment: The recruitment date range (month and year)

Authors response: Recruitment of participants was done from 19th July, 2019 through to 23rd July, 2019. 

Reviewers comment: A description of any inclusion/exclusion criteria that were applied to participant recruitment,

Authors response: This has been reviewed in the methods section as follows;

Inclusion and Exclusion criteria

The study inclusion criteria was households keeping livestock in the selected villages and those whose household head or any person above the age of 18 year was available. The exclusion criteria for the household survey were those households in which the respondents were eligible for the interview but could not able to provide information required due to inability to communicate, keeps missing scheduled appointments and providing inaccurate information during data collection. 

Reviewers comment: A table of relevant demographic details

Authors response: This has been included in the methods section narrative

Reviewers comment: a statement as to whether your sample can be considered representative of a larger population, 

Authors response: This has been addressed in the methods section

Reviewers comment: a description of how participants were recruited 

Authors response: This had been described in the methods section using narrative

Reviewers comment: descriptions of where participants were recruited and where the research took place.

Authors response: This had been described in the methods section using narrative

5. Review Comments to the Author

Comments Reviewer #1: Thank you for the opportunity to read this interesting article. This is a very clearly written piece of work and offers a useful summary of a recent Anthrax outbreak in western Kenya. I have been invited to review this article as a social scientist who has worked in this region. My comments primarily pertain to the KAP findings and discussion. I am not an expert on Anthrax.

In terms of the background information, it would be helpful to know if the village in question is rural or peri-urban, given the vast differences in how people live in Nyalenda and Manyatta compared to other parts of Kisumu East sub-county. Given the geographical differences across the sub-county, why is it that the KAP respondents were taken from all five wards rather than just those which shared a similar make-up to the village location?

Authors response: We found out that although there were two major geographical regions, livestock keeping was practiced in all the five wards so they share the same characteristics in terms of livestock keeping.

Comments Reviewer #1: 

The term ‘knowledge’ and ‘lack of knowledge’ are used in ways that are problematic from a social science perspective. There are a few different issues here. First, it is not clear what questions about Anthrax were asked of respondents. Presumably these related to modes of transmission, infection and prevention? Should this questionnaire not be included as an appendix to the article? The article should make clear that participants are being questioned on ‘biomedical understandings of modes of transmission, infection, and transmission’. It may be commonplace in biomedical publications but to gloss this as ‘knowledge about Anthrax’, but for social scientists this is unhelpful and problematic, because it undervalues other kinds of knowledge and insight that people may have about Anthrax (or aremo, or other related cattle diseases/disorders) and dismisses local knowledge as irrelevant, and as mere ‘ignorance’.

Authors response: 

We have revised The term ‘knowledge’ and ‘lack of knowledge’ to Poor knowledge and good knowledge after we realised that all respondents must have some level of knowledge. There were specific set of questions related to cause, transmission and prevention. We will provide the questionnaire in the appendix.

Comments Reviewer #1: 

In the first quotation, you where you use the example of Aremo, it is not clear if you are giving this as an example of knowledge, or ‘lack of knowledge’. What this example does prove very nicely, is the comment in your conclusion, that previous experience is very important for contructions of knowledge, partly because the symptoms are very dramatic. Rather than just using this quote as ‘an example of knowledge’, it would be better to break up the concept of knowledge and use this quote as an example of a specific way in which knowledge is obtained. This is very helpful to know from a public health perspective, because it suggests that using images of infected cattle could be a helpful tool for educating people about the disease.

Authors response: 

This was to depict knowledge and to explain that the moment you have a chance of seeing an animal presenting with clinical symptoms of anthrax, you might not forget how it looks like due to its dramatic nature.

Comments Reviewer #1: 

On Aliya, I wondered if you could say more about the gendered dimensions of risk here. You mention the risk of anthrax remaining in the meat despite the drying process, but not the possible risk of the preparation process. Is this a female activity? The same goes for skinning the animal. Is it more like that a man would do this? Or are these unlikely to be modes of transmission?

Authors response: 

There are two main instances for disease transmission which is gendered. The first one is exposure by skinning with high risk being among male gender and preparation while cooking with risk being high in female gender at this point.

Comments Reviewer #1: In general the article presentations quotations as examples of a form of knowledge, an attitude or a practice, but it doesn't always make an argument with those presentations. It is not always clear enough why the particular examples have been chosen, and what message the authors want the reader to take form the presentation of these specific examples.

Personally, I do not see much difference between ‘attitudes’ and ‘practices’ as you present them, given that both sections appear to describe what people would do if they encountered meat from a dead animal, and wonder if these two sections would be better combined.

Authors response: We have attempted to revise the sections on ‘attitudes’ and ‘practices’ to present attitude as a way of thinking and practices as putting thinking into action. 

Comments Reviewer #1: In the section where you discuss factors that increase biomedical understandings of Anthrax, why not talk about the factors that don’t have any impact? For example, it is very interesting that gender does not appear to shape access to this kinds of knowledge. Why do you think this is?

Authors response: There are no differences in gender, education level and occupation variations between the rural and peri-urban settlement resulting into lack of association of these dependent variables with knowledge of anthrax.

Comments Reviewer #1: In the conclusion, I would have liked to know specifically how a one health approach would have helped here specifically. Also, in what sense does the study confirm and contradict other work? Would you expect things to be similar in Sweden anyway in terms of understanding of aetiologies and engagement in modes of prevention? These comments are left hanging a little.

Authors response: We have reviewed the section to add more information on one health concept

Comments Reviewer #2: while the manuscript has technically presented a sound piece of scientific research, one cannot be certain that statistical analysis been performed appropriately and rigorously. The data that computed level of knowledge for the

community members interviewed on cause transmission, symptoms and prevention need to be illustrated for transparency e.g. as a supplementary text. The manuscript also requires major language edits and appropriate referencing of literature.

Authors response: The authors will provide the data as Ms. Excel spreadsheet

---

## [Editor Report · Decision Letter 1]

28 Jul 2021

PONE-D-20-34615R1

‘We dry contaminated meat to make it safe’: An Assessment of Knowledge, Attitude and Practices on Anthrax during an outbreak, Kisumu, Kenya, 2019

PLOS ONE

Dear Dr. Mugo,

Thank you for submitting your manuscript to PLOS ONE. After careful consideration, we feel that it has merit but does not fully meet PLOS ONE’s publication criteria as it currently stands. Therefore, we invite you to submit a revised version of the manuscript that addresses the points raised during the review process.

Before this can be accepted, please ensure it is copyedited.

We look forward to receiving your revised manuscript.

Kind regards,

Eric Fèvre

Academic Editor

PLOS ONE

Journal Requirements:

Additional Editor Comments (if provided):

Thank you for your response to the reviews. The copy editing issue remains in the new manuscript and it would be valuable to have this manuscript reviewed by a copy editor
---

## [Author Response · Author response to Decision Letter 1]

2 Oct 2021

Line 53; Article ‘the’ to read The household anthrax. 

Line 54 putting article ‘the’ 

Line 60 Article ‘the’ 

Line 61 Article ‘the’ and adding ‘was’ 

Line 63 Article ‘the’knowledge’ 

Line 69 Article ‘the’ 

Line 72 ‘were’ instead of ‘was’ 

Line 126 ‘demonstrating’ instead of ‘demonstration’ 

Line 142 ‘Before’ instead of ‘prior’ 

Line 218 ‘were’ instead of ‘was’ 

Line 275 Article ‘the’ 

Line 277 inserted ‘with the’ 

Line 301 article ‘the’

Line 305 replaced ‘aged’ with ‘elderly ‘

Line 350 removed ‘having’ 

Line 351 removed ‘only’ 

Line 370 replaced ‘usually’ with ‘which is’ 

Line 378 inserted article ‘the’ 

Line 383 inserted article ‘the’ 

Line 384 inserted article ‘the’ 

Line 386 revised changing ‘sort’ to ‘sought’ 

Line 409, 410 and 411 revised to capture more details on previous anthrax outbreaks in Kisumu 

Line 418 inserted article ‘The’ 

Line 429 inserted article ‘The’ The existence 

Line 434 replaced ‘food borne’ with ‘food-borne’ 

Line 441 inserted article ‘The’ 

Line 445 inserted article ‘The’ 

The limitations section was revised to add the confirmatory diagnosis was done in line with guidelines 

Line 480 replaced ‘concerning’ with ‘in relation’

---

## [Editor Report · Decision Letter 2]

12 Oct 2021

‘We dry contaminated meat to make it safe’: An Assessment of Knowledge, Attitude and Practices on Anthrax during an outbreak, Kisumu, Kenya, 2019

PONE-D-20-34615R2

Dear Dr. Mugo,

We’re pleased to inform you that your manuscript has been judged scientifically suitable for publication and will be formally accepted for publication once it meets all outstanding technical requirements.

Kind regards,

Eric Fèvre

Academic Editor

PLOS ONE

Additional Editor Comments (optional):

Many thanks to the authors for the revisions, which have resulted in a much improved manuscript.
---

## [Editor Report · Acceptance letter]

18 Oct 2021

PONE-D-20-34615R2 

‘We dry contaminated meat to make it safe’: An Assessment of Knowledge, Attitude and Practices on Anthrax during an outbreak, Kisumu, Kenya, 2019 

Dear Dr. Mugo:

I'm pleased to inform you that your manuscript has been deemed suitable for publication in PLOS ONE. Congratulations! Your manuscript is now with our production department. 

Kind regards, 

on behalf of

Prof. Eric Fèvre 

Academic Editor

PLOS ONE